# Dietary Habits in Early Pregnancy in a Multi-Ethnic Population: Results from the PROMOTE Cohort Study

**DOI:** 10.3390/nu17233729

**Published:** 2025-11-27

**Authors:** Ania (Lucewicz) Samarawickrama, James Elhindi, Yoon Ji Jina Rhou, Sarah J. Melov, Justin McNab, Mark McLean, Ngai Wah Cheung, Ben J. Smith, Tim Usherwood, Victoria M. Flood, Dharmintra Pasupathy

**Affiliations:** 1Reproduction and Perinatal Centre, Faculty of Medicine and Health, The University of Sydney, Westmead, Sydney 2145, Australia; 2General Practice Clinical School, Faculty of Medicine and Health, The University of Sydney, Camperdown 2006, Australia; 3Specialty of Medicine, Faculty of Medicine and Health, The University of Sydney, Camperdown 2006, Australia; 4Department of Diabetes & Endocrinology, Westmead Hospital, Sydney 2145, Australia; 5Westmead Institute for Maternal and Fetal Medicine, Westmead, Sydney 2145, Australia; 6Sydney School of Public Health, The University of Sydney, Camperdown 2006, Australia; 7Westmead Institute for Medical Research, Westmead, Sydney 2145, Australia; 8George Institute for Global Health, Sydney 2000, Australia; 9University Centre for Rural Health, Northern Rivers Rural Clinical School, The University of Sydney, Lismore 2480, Australia; 10Specialty of Obstetrics, Gynaecology and Neonatology, Westmead Clinical School, Faculty of Medicine and Health, The University of Sydney, Westmead, Sydney 2145, Australia

**Keywords:** obstetrics, diet, nutrition, gestational diabetes, lifestyle risk factors, cardiometabolic health

## Abstract

Introduction: The PROMOTE cohort study is a prospective pregnancy cohort study that seeks to improve the understanding of cardiometabolic risk and determinants, such as diet, during pregnancy in a multi-ethnic population. Increasing age and obesity has resulted in an increased risk of cardiometabolic complications during pregnancy, including gestational diabetes. Trials of lifestyle interventions have so far produced mixed results, partly due to a wide variation in the methods, duration, adherence and type of dietary intervention. There is a need for high quality data about dietary habits in pregnancy, particularly in multi-ethnic populations. Objectives: In this study, we report the dietary habits of women in early pregnancy in the population of interest. We report early data seeking to assess the relationship between dietary patterns and risks of gestational diabetes. Methods and analysis: The PROMOTE cohort study is a prospective pregnancy cohort study recruiting pregnant participants with <16 weeks gestation in an area of high social and cultural diversity in western Sydney, Australia. The participants are surveyed about their physical activity levels, diet quality, emotional wellbeing and sociodemographic status using validated tools. Participants have consented to the use of routinely collected clinical and social data, including medical conditions, body mass index (BMI), blood pressure (BP) and glycaemia. The follow-up is from routinely collected data. This paper presents dietary data. Results: A total of 459 participants were recruited (*n* = 459), including 416 with GDM data available, at the conclusion of the first 2 years of recruitment. No participants met national dietary guideline recommendations. Fifty-six participants (*n* = 56, 13%) met a pragmatic composite standard of favourable diet, defined as two servings of vegetables and two servings of fruit per day, with a maximum of one discretionary serving per day. Over half the participants (*n* = 215, 51%) reported an adequate daily fruit intake. In total, 7 participants ate at least five servings of vegetables per day (*n* = 7, 1.7%), 61 participants (14.7%) ate three or more servings of vegetables per day and 212 (51.2%) participants reported one discretionary item per day. The data suggest that few women meet dietary recommendations in pregnancy. The association between dietary habits and GDM was unable to be assessed. The study was underpowered to detect an association due to the highly skewed distribution of dietary patterns in our population. **Conclusions:** The uptake of dietary recommendations was very low in our sample. This represents a major population health concern. Multi-level approaches are urgently needed to address poor dietary habits in pregnancy.

## 1. Introduction

### 1.1. Background/Rationale

Maternal dietary habits are widely discussed as a modifiable risk factor for cardiometabolic complications in pregnancy, including gestational diabetes (GDM) [1,2,3,4]. National and international guidance suggests that pregnant women follow recommendations regarding dietary intake in pregnancy, such as a certain number of servings of fruits, vegetables, meat or alternatives and dairy products per day, limiting discretionary or ‘junk food’ items while balancing energy intake with need [5,6]. At the same time, both the academic literature and guidance widely cite obesity and excessive gestational weight gain as a risk factor for cardiometabolic complications during pregnancy, including gestational diabetes [7,8,9].

However, despite observational evidence and clinical guidance recommending the benefits of favourable diets during pregnancy, lifestyle intervention trials during pregnancy aiming to reduce risks and consequences of gestation diabetes by way of diet have produced mixed outcomes, with the subsequent proliferation of a wide variety of literature reviews, umbrella reviews and systematic reviews aiming to make sense of the variations in study population, intervention length, duration and design [10,11,12,13,14,15]. The best way to measure, monitor and address dietary exposures and their determinants remains in doubt.

At the same time, despite clear observational evidence and guidance, the uptake of dietary recommendations is low [16]. Individual and social determinants of adherence to guidelines remains poorly defined in the academic literature [17]. There is therefore a need to identify the factors associated with adherence or otherwise to dietary recommendations—at both a personal and systemic level.

The social and cultural determinants of diet remain poorly understood, with of the importance of reflecting ethnic and cultural diversity in clinical trials gaining increasing attention [18,19,20]. There is a need to widen the scope of the existing literature and improve the inclusion and study of migrant and diverse populations.

### 1.2. Objectives

This paper presents the results from the first two years of the PROMOTE cohort study. The objectives are to report on early pregnancy dietary habits in a multi-ethnic community and the associated maternal characteristics. Additionally, this study aims to report early data seeking to assess the association between dietary intake and gestational diabetes.

## 2. Methods

### 2.1. Overall Study Design and Population

A detailed protocol outlining the methods of the PROMOTE cohort study has been published [21]. Briefly, women are offered recruitment in early pregnancy (<16 weeks gestation). Participants consent to the use of routinely collected clinical data, including clinical history, baseline and longitudinal measurements, such as body mass index and blood pressure, and pathology results including oral glucose tolerance test (OGTT) results. Additionally, as described in the protocol, participants are requested to complete surveys about their diet, physical activity levels, sociodemographic circumstances and mental health, and are optionally offered biobanking of maternal and cord blood samples [21]. The PROMOTE cohort study is dynamic, with sub-studies and extension studies being included. In keeping with the published protocol, data from participants recruited between February 2022 and February 2024 were analysed, with data regarding dietary habits being the focus of this reporting.

### 2.2. Data Collection for Baseline Characteristics

The baseline characteristics of the study population have been published elsewhere, and include data extracted from clinical records (such as demographics, medical and obstetric history, BMI and blood pressure) [21]. Mental health screening was performed in routine clinical practice, as described in the protocol, by way of the Edinburgh Perinatal Depression Scale (EPDS) [22] and additionally by the Depression Anxiety and Stress Scale (DASS21) [23]. Additionally, participants were asked about additional sociodemographic characteristics, including household structure, education level, employment status and income (Appendix B). Additional sociodemographic characteristics were extracted from routine clinical records (Appendix C). These included age, religion, marital status, postcode, local health district, country of birth, year of arrival in Australia, ethnicity and identification as Aboriginal or Torres Strait Islander. The Socio-Economic Indexes for Areas (SEIFA) score was determined through the existing clinical record, which links the participant’s address to an area estimate of socio-economic advantage and disadvantage [24]. The SEIFA score is informed by Australian national census data and is informed by four subscores: the Index of Relative Socio-economic Advantage and Disadvantage, the Index of Relative Socio-economic Disadvantage, the Index of Education and Occupation and the Index of Economic Resources [24].

### 2.3. Data Collection for Dietary Quality

Maternal dietary intake was measured using a short food frequency questionnaire (FFQ) adapted from the Centre for Epidemiology and Evidence, NSW Ministry of Health [25]. This is included in Appendix D. The questionnaire consists of 13 short recall questions designed to broadly measure dietary intake by food group, including servings of fruit, vegetables, bread, breakfast cereal, dairy, pasta/rice/cooked cereals, legumes, meats/fish/eggs, processed meats, discretionary items, soft drinks/juices and takeaway foods. Use of a short FFQ was chosen pragmatically—short FFQs are amenable to implementation in a clinical setting, as they are completed within minutes, compared to the use of longer instruments such as weighted food diaries or longer recall questionnaires. As ours is a pragmatic cohort study implemented in a busy clinical setting, the use of a short screener was chosen. The FFQ therefore is able to distinguish between participants, but is not designed to answer more granular and detailed nutritional questions like micronutrient or energy intake.

Dietary characteristics across the population were reported as binary variables across domains for meeting dietary requirements as follows: yes/no for fruit intake, vegetable intake, a combined category including bread, breakfast cereal, pasta/rice and cooked cereal, a combined category of discretionary foods including items like processed meat products, hot chips, crisps, soft drinks and takeaways, meat and an alternative food category. Furthermore, a composite measure of ‘good’ diet was informed by Australian dietary guidelines, which recommend the following daily servings during pregnancy: 5 vegetables, 2 fruit, 3.5 meat or alternatives, 2.5 dairy and a limited intake of foods with high saturated fats, salt or added sugars [5,26]. Acknowledging methodological challenges [27], the low uptake of guidance and the need to acknowledge the combined influence of multiple poor dietary exposures [28,29], a pragmatic compromise was made to define a favourable diet by population quartiles for the following domains: fruit, vegetables and discretionary items.

Challenges with both a priori and posteriori approaches to dietary assessment are well documented, with the use of both approaches widely cited in the existing literature in nutritional epidemiology [30,31,32,33,34,35]. Therefore, the decision was made to report both adherence to pre-defined dietary guidance as well as the pragmatic compromise by population-defined quartiles, taking the top quartile for vegetable intake, top quartile for fruit intake and top quartile for discretionary items. In this population, this became defined as two servings of fruit plus two servings of vegetables and a maximum of one serving of discretionary items daily.

### 2.4. Data Collection for Physical Activity

Physical activity levels were measured using the Active Australia Survey [36], administered at recruitment. The AAS consists of eight questions designed to measure participation in leisure-time physical activity and five statements to assess knowledge about health benefits of physical activity. Further details of the methodology have been published elsewhere [21].

### 2.5. Data Collection for Gestational Diabetes

Data around the incidence of gestational diabetes were extracted from existing clinical records, in keeping with the published protocol [21]. All participants were offered universal screening for GDM as per local standard practices. This includes a standard 2 h 75 g oral glucose tolerance test (OGTT) at 24–28 weeks gestation, with some women also being offered an early OGTT on the basis of risk factors. GDM was treated as a binary outcome using the Australian Diabetes in Pregnancy Society (ADIPS) criteria [37] for the diagnosis of GDM: a diagnosis of GDM is made if one or more of the following glucose levels are elevated: fasting ≥ 5.1 mmol/L, 1 h glucose ≥ 10.0 mmol/L and 2 h glucose ≥ 8.5 mmol/L. Additionally, absolute venous glucose results at 0, 1 and 2 h were also extracted from the clinical record. Those without GDM or OGTT results available were excluded from analysis.

### 2.6. Study Size

We report here the analysis of the first two years of recruitment in the PROMOTE cohort study, with an emphasis on dietary data. Recruitment is ongoing.

### 2.7. Statistical Methods

Statistical analysis was conducted in R Studio Version 4, as per the protocol. Hypotheses were conducted at a family-wise significance level of 0.05 with a two-sided alternative. Appropriate Bonferroni corrections were made to the significance level of individual hypotheses. Summary statistics describe the characteristics of the cohort. Descriptive statistics are presented on the relationship between outcomes of interest and baseline risk factors. Continuous variables are summarised as mean ± standard deviation if normally distributed and median and interquartile range if not. Categorical variables are presented as frequency (%) in relevant categories.

### 2.8. Ethics

The study has been approved by the Western Sydney Local Health District Human Research Ethics Committee (2021/ETH00287).

## 3. Results

### 3.1. Participants

The participant characteristics have been published elsewhere [38]. A short summary is reproduced here. In the period of March 2022—the end of February 2024, of 582 potential participants, 507 (87%) agreed to participate. Those who elected to withdraw (*n* = 7), those expecting twins (*n* = 9), those who were unable to be matched to a clinical record (*n* = 6) and those with pre-existing diabetes (*n* = 26) were excluded, leaving a total of 459 participants for analysis. See Figure 1 for the CONSORT diagram (reproduction from prior publication) [38].

### 3.2. Baseline and Sociodemographic Characteristics

The participant baseline and sociodemographic characteristics are published elsewhere [38]. A summary of this data is reproduced here in Table 1 [38]. The mean age was 32.48 (range 21–43), the mean BMI was 25.3 kg/m^2^ (range: 16.8–50.9), the majority of participants (*n* = 316; 69%) were born outside Australia, with India (*n* = 112; 24.5% of total cohort), Pakistan (*n* = 22; 4.8%), Nepal (*n* = 22; 4.8%), China (*n* = 26; 5.7%) and Afghanistan (*n* = 18; 3.9%) being the most common countries of birth. Self-reported ethnicities included South Asian (*n* = 172; 37%), White (*n* = 90; 20%), Middle Eastern (*n* = 81; 18%) and South-East Asian (*n* = 77; 17%). Three participants identified as Aboriginal or Torres Strait Islander (0.65%).

Socio-Economic Indexes for Areas scores were available for 351 participants and are published elsewhere [38]. This is also summarised in Appendix A. The household income in Australian dollars was available for 416 participants and reported in categories, with the following number of participants in each bracket, <AUD 50,000 (*n* = 23; 5%), AUD 50–100,000 (*n* = 96; 21%), AUD 100–200,000 (*n* = 153; 33%) and >AUD 200,000 (*n* = 75; 16.5%), while 107 (23.5%) participants selected ‘Don’t know/Prefer not to say’. Education levels were as follows: incomplete high school (*n* = 10; 2%), completed high school (*n* = 38; 8.3%), completion of high school plus a post-school qualification such as a certificate/diploma (*n* = 82; 18%) and completion of high school plus a university qualification (*n* = 317; 69%). Nine (1.9%) participants reported smoking and three (0.6%) reported illicit drug use. No participants reported alcohol use.

### 3.3. Dietary Characteristics Across the Population

The descriptive statistics for dietary intake by food group are summarised in Table 2. Dietary characteristics across the studied population were reported as a binary variable across categorised domains. Amongst the 416 participants, the following fruit and vegetable intake was reported: a fruit intake of two servings or more per day (*n* = 215; 51.7%), a vegetable intake of five or more servings per day (*n* = 7, 1.7%) and a vegetable intake of three or more servings per day (*n* = 61; 14.7%). With respect to discretionary items, the following was reported: no discretionary items (*n* = 12, 2.9%) and one discretionary item per day (*n* = 212; 51%). Foods with predominant carbohydrates (breads and cereals) were reported as 8.5 or more per day (*n* = 3, 0.7%) and 2 or fewer per day (*n* = 117; 28%). Meats and alternatives were reported as follows: 3.5 servings or more per day (*n* = 2; 0.5%) and 2 or more servings per day (*n* = 32; 7.7%). Dairy intake was reported as follows: 3.5 or more servings per day (*n* = 6; 1.4%) and 2 or more servings per day (*n* = 137; 32.9%). Frequency histograms of dietary intake are found in Appendix A.

The composite assessment of diet is reported in two ways: the first being those who meet the national guideline recommendations, and the second being those who meet the alternative definition of a favourable diet, for the purposes of this study. The national guideline recommendations suggest five or more vegetable servings per day and two or more fruit servings per day, and to limit discretionary items. No participants met these recommendations (*n* = 0, 0%). In the alternative composite measure for this study, defined as a ‘favourable diet’, comprising two or more servings vegetable servings per day, two or more fruit servings per day and a maximum of one serving of discretionary food choices, 13% women reported meeting those servings (*n* = 56).

### 3.4. Characteristics Associated with Dietary Behaviours

A pragmatic composite definition of a ‘favourable’ diet consisting of two or more vegetable servings per day plus two or more fruit servings per day and a maximum of one discretionary item per day was analysed. The assessment of demographic characteristics associated with a ‘favourable diet’ are presented in Table 3, with clinical characteristics in Table 4.

Multiple demographic characteristics were assessed in relation to dietary behaviour, including maternal age, household structure, maternal education level, household income, financial autonomy and self-reported ethnicity. No demographic characteristics were clearly associated with dietary behaviours, although there was a possible trend towards maternal older age being associated with a more favourable dietary habit, with the mean age amongst those with a favourable diet being 34 years (range 31–38) and those with an unfavourable diet being 32 years (range 29–36); *p* = 0.02. There was also the suggestion of an association between a higher educational attainment and an increased likelihood of a favourable diet, with university level educational attainment reported by 80% of those with a favourable diet (*n* = 43) and 69% (*n* = 242) of those with an unfavourable diet. There was no statistically significant association between self-reported ethnicity and a favourable diet.

The clinical characteristics assessed for associations with reported dietary habits included maternal BMI, maternal obstetric history, including the history of gestational diabetes, history of assisted conception, breastfeeding history and history of polycystic ovarian syndrome (PCOS), and the history of recurrent miscarriage.

Analysis of single time point mental health screening scores, including the DASS21, did not detect a statistically significant association between depression, anxiety or stress levels and dietary habits. Screening by way of the EDPS also did not demonstrate a statistically significant association between maternal distress levels and dietary habits. These results are summarised in Table 5.

### 3.5. Overlap with Physical Activity Behaviours

Physical activity behaviours were analysed by category and intensity and are summarised in Table 6. There was no statistically significant association found between physical activity behaviours and dietary habits. Sufficient physical activity levels were reported by 48% of those with a favourable dietary habit (*n* = 27, 48%) compared to 38% of those with an unfavourable dietary habit (*n* = 136, 38%); *p* = 0.29.

### 3.6. Dietary Habit and Rate of Gestational Diabetes

Of the 416 participants, 104 (25%) had GDM and 312 (75%) did not have GDM. Amongst the 25 participants with a favourable dietary habit, the GDM rate was 28% (*n* = 7, 28%). Amongst the 391 participants with an unfavourable dietary habit, the GDM rate was 25% (*n* = 97, 25%); *p* = 0.81. The characteristics of those with and without GDM are summarised in Appendix A.

### 3.7. Dietary Habit and OGTT Results

Oral glucose tolerance test (OGTT) data were available for 379 (82.5%) participants. The results are summarised in Table 7. There was no statistically significant association detected between composite dietary habits (favourable/unfavourable) and glucose levels.

### 3.8. Dietary Intake by Food Group and Gestational Diabetes Rate

The dietary intake of food groups was analysed for associations with gestational diabetes rates. No statistically significant association was found between individual dietary food groups and the rate of gestational diabetes, with the exception of an analysis of discretionary items. The gestational diabetes rate amongst those with a reported intake of zero discretionary items per day was 58% (*n* = 7) and amongst those with a reported intake greater than this was 24% (*n* = 97); *p* = 0.01. The interpretation of this result is limited by the extremely small sample size in the group reporting the zero discretionary items (being seven participants) and precludes a reasonable interpretation. The results are summarised in Table 8.

### 3.9. Indicative Sample Size Needed to Assess Association Between Dietary Intake and Gestational Diabetes

The feasibility of sample size calculation was examined. Of the 416 participants, 104 (25%) had GDM and 312 (75%) did not have GDM. Amongst the 25 participants with a favourable dietary pattern, the GDM rate was 28% (*n* = 7, 28%). Amongst the 391 participants with an unfavourable dietary pattern, the GDM rate was 25% (*n* = 97, 25%); *p* = 0.81—i.e., comparing a GDM rate of 7/25 (28%) against 97/391 (25%). As the GDM rate between groups was near even—and the rate was actually higher for those with a favourable diet, raising question about the adequacy of small sample size in the favourable diet group and resultant skewed distribution—a sample size calculation was unable to be meaningfully performed.

## 4. Discussion

The PROMOTE cohort study captures cross-sectional lifestyle exposures in a highly ethnically and socio-economically diverse setting of western Sydney and seeks to address a need for high quality data about diverse populations.

Our study has observed that the uptake of dietary recommendations during pregnancy is poor in our population. National data are limited but suggest a similarly low uptake of dietary recommendations in the general population, with the Australian Institute of Health and Welfare reporting that, in 2022, 56% of Australian adults did not meet fruit intake recommendations, while 94% of adults did not meet vegetable recommendations [39]. Importantly, there is no current standard for the national reporting of dietary habits during pregnancy, even though other cardiometabolic risk factors like BMI are routinely nationally collected and reported [40]. The implementation of a comprehensive national strategy to address the harms of adverse dietary exposures and monitor the reach and response to interventions is likely to remain challenging in the absence of robust reporting of dietary intake, alongside other cardiometabolic risk factors like obesity, physical activity levels and background clinical conditions.

Our study has observed that our contemporary Australian pregnant population has multiple pre-existing subclinical cardiometabolic risk factors, including a high prevalence of overweight and obesity. It is not yet clear as to whether uniform, population-wide dietary recommendations in pregnancy should be applied to those with significant or multiple subclinical risk factors, such as obesity, a history of PCOS, a significant family history or ethnic-specific risk. Future studies need to be powered not only to detect a difference between the gestational diabetes rate and an adherence to population-level general recommendations, but also to assess the value of dietary habits tailored to risk factor subgroups. The current national guidance makes general recommendations at a population level, with minimal amendments to address the variations in subclinical risk present in the population [5,26]. It is possible that individualised dietary recommendations may be needed to adequately address underlying subclinical cardiometabolic risk, as it varied across the population.

Our study observed a potential association between a slightly older age and an increased likelihood of a favourable diet, as well as a possible trend towards a favourable diet amongst those with a higher education level. Identifying social or demographic subgroups at risk of GDM may have implications for the type and tailoring of health interventions varying by health literacy.

The dietary composition in our data is highly skewed. This has major implications for the sample size needed to detect a clinically meaningful difference in GDM between dietary groups, especially when adjusting for co-occurring risk factors for GDM like BMI, ethnicity or clinical history. Our study was unable to meaningfully calculate a sample size adequate to answer these questions due to a highly skewed distribution. Any future power calculation would also need to address the need to control for multiple co-exposures and to analyse these co-occurring lifestyle exposures both separately and together, such as diet and physical activity levels. These factors make the calculation of a sample size in a skewed population challenging.

The implementation of dietary guidelines brings with it major challenges with adherence [41]. The low uptake of dietary recommendations in our population raises questions about the practicality of current recommendations and about whether a pragmatic approach to dietary improvement, focusing on incremental improvements, would still yield clinical benefits.

One weakness of the study is the use of a single time point measurement of dietary habits. The duration of dietary interventions required to result in meaningful differences in cardiometabolic risk profiles and gestational diabetes rates is unknown. The existing reviews have noted a wide variation in the timing and duration of reported interventions [42]. Short-term changes to diet may not yield clinical benefits, so future studies should be designed where possible and practical to measure dietary exposures over much longer periods, ideally from pre-conception.

Another limitation of this study is the use of a short food survey, which does not include a comprehensive list of items. The logistical implementation of dietary assessment remains a challenge, with the practicality of short food questionnaires being weighed against the more cumbersome food frequency questionnaires. While all food recall methods are at risk of an under-recall bias [43], emerging methods of data collection include the smart-phone enabled AI detection of foot intake by photographs, and may provide a way forward, balancing practicality and detail [44]. The benefits and limitations of a range of ways of measuring dietary intake (such as screening questionaries, food frequency questionnaires, dietary recalls, weighted food record, etc.) are well documented [31,34,45], as well as the challenges inherent in assessing and reporting adherence [32]. Our study has adopted a pragmatic approach to dietary assessment, with a brief food frequency questionnaire designed to distinguish between participants, but not able to provide a highly granular detailed dietary assessment. Emerging tools, such as short screeners like the FiGO questionnaire, have also been compared to more detailed dietary assessment tools and appear to have utility [46].

Furthermore, the use of pragmatic approaches to distinguish so-called ‘favourable’ and ‘unfavourable’ dietary habits also has its limitations. The existing literature describes the benefits and limitations of the use of pragmatic, population-based cut-offs in dietary epidemiology [30,34,35]. Furthermore, the existing cohort studies in non-pregnant populations have also made use of pragmatic cutoffs, including the use of median cut-offs, tertials, quartiles and other variations [47,48,49,50]. The utility of these lies in their discriminatory power, but they are difficult to reproduce across populations.

In response to some of these limitations, proposals to develop more universally acceptable dietary indices for use in diverse settings have arisen, such as the development of a Global Diet Quality Score (GDQS) [51]. There is work in progress seeking to validate and assess the utility of such scores for pregnant populations, and the outcome of such work will enhance the study of maternal nutrition globally [33,52].

Finally, our null findings should be interpreted with caution due to the limitations inherent to FFQs, including recall bias, portion misinterpretation and the so-called ‘flattened slope phenomenon’, with under-reporting and over-reporting at the high and low ends of intake; these are major concerns to consider when making sense of null findings [53]. Whether novel statistical approaches or biomarker analysis can address some of these shortcomings remains in need of exploration [54,55,56].

Our data indicate a low uptake of dietary recommendations and the need for urgent action to address barriers and support change. Most of the published literature about lifestyle interventions for gestational diabetes presents individual-level interventions provided within a clinical service or behaviour-change interventions only reporting on impacts on health behaviours rather than clinical outcomes [11,14]. There remains significant scope to widen the variety and scope of interventions, such as direct-to-consumer or digital-first interventions, place-based interventions (such as partnerships with existing community recreational spaces), family-based interventions, social prescribing interventions or culturally adapted interventions. Finally, interventions focussed on underlying determinants of food intake, such as food insecurity and cooking skills, have received recent attention and warrant further investigation [17,57].

Despite its limitations, this study provides insights into the dietary habits in a multi-ethnic urban population and highlights the urgent need for action.

## 5. Conclusions

While pregnancy is often discussed as a potential window of opportunity to improve lifestyle exposures for the benefit of long term cardiometabolic health, our study suggests that suboptimal dietary habits remain a major adverse lifestyle exposure in pregnancy, including in ethnically diverse and migrant populations. Our study highlights the need for urgent action. Significant methodological challenges need to be tackled in order to meaningfully assess dietary intake, including a sample size well powered to detect a difference between dietary groups.

## Figures and Tables

**Figure 1 nutrients-17-03729-f001:**
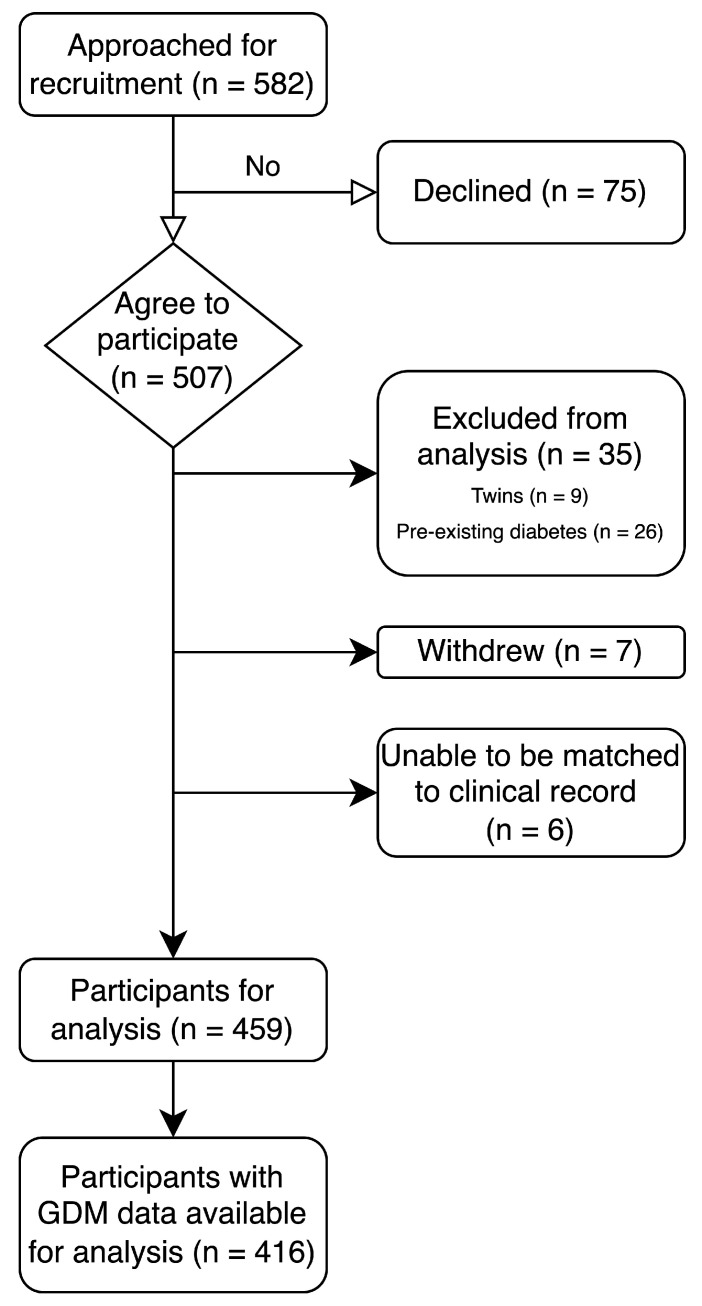
CONSORT diagram.

**Table 1 nutrients-17-03729-t001:** Descriptive characteristics of participants [38].

Maternal		All Participants (N = 459)	Participants with GDM Data Available (N = 416)
	Age, median (range)	33 (21–43)	32 (21–43)
	20–24 years, *n* (%)	21 (4.5)	22 (5)
	25–34 years, *n* (%)	265 (58)	252 (61)
	35–39 years, *n* (%)	143 (31)	119 (29)
	40+ years, *n* (%)	30 (6.5)	21 (5)
	Parity, *n* (%)		
	Para 0	140 (30.5)	120 (29)
	Para 1	217 (47)	199 (48)
	Para 2	68 (15)	65 (16)
	Para 3	26 (6)	25 (6)
	Para 4+	8 (1.7)	7 (1.7)
	BMI kg/m^2^, mean (range)	25.3 (16.8–50.9)	26.5 (16.8–50.9)
	≤18.5	10 (2)	8 (2)
	18.5–25	211 (45)	195 (47)
	25–30	138 (30)	127 (31)
	≥30	100 (22)	86 (21)
	Smoking status, *n* (%)	9 (2)	9 (2)
	Reported alcohol use in pregnancy, *n* (%)	0 (0)	0 (0)
	Medical history		
	Hypertension, *n* (%)	7 (1.5)	6 (1.4)
	Diabetes	Excluded	Excluded
**Sociodemographic**			
	Self-reported ethnicity, *n* (%)		
	Middle Eastern	81 (18)	79 (19)
	South-East Asian	77 (17)	69 (16.5)
	South Asian	172 (37)	153 (37)
	White	90 (20)	79 (19)
	Other	39 (8.5)	36 (8.7)
	Country of birth, *n* (%)		
	Australia	143 (31)	131 (33)
	Overseas	316 (69)	285 (67)
	Number of people in household, median (range)	3 (1–12)	3 (1–12)
	Number of children in household, median (range)	1 (0–12)	1 (0–12)
	Education level completed, *n* (%)	Available for 457	Available for 414
	Incomplete high school	10 (2)	10 (2.4)
	Completed high school	38 (8.3)	36 (8.7)
	TAFE * certificate/diploma	82 (18)	74 (18)
	University/tertiary institute	317 (69)	285 (68)
	Other	4 (0.8)	3 (0.7)
	Prefer not to say/did not answer	6 (1.3)	6 (1.4)
	Household income bracket AUD, *n* (%)	Available for 454	Available for 411
	<50,000	23 (5)	22 (5)
	50–100,000	96 (21)	85 (20)
	100–200,000	153 (33)	139 (33)
	>200,000	75 (16.5)	64 (15)
	Don’t know/prefer not to say	107 (23.5)	101 (24)
	Missing	12	12
**Pregnancy**			
	Gestational age at recorded first visit in weeks, mean (range)	6.2 (5–13)	6.2 (5–13)
	Mode of conception, *n* (%)		
	Spontaneous	435 (95)	395 (94)
	IVF	20 (4.4)	17 (4)
	Ovulation induction	3 (0.6)	3 (0.7)
	Natural fertility services	1 (0.2)	1 (0.2)
	Edinburgh perinatal depression screening score, *n* (%)	Available for 455	Available for 412
	Score range 0–9	406 (89)	371 (90)
	Score range 10–12	31 (7)	27 (6.5)
	Score range ≥ 13 or response to Q10 on self-harm	18 (4)	14 (3.4)
	History of GDM, *n* (%)	63 (18) ^#^	54 (18) ^#^

* TAFE—Technical and Further Education, a vocational and training system in Australia with a focus on post-school vocational qualifications at certificate and diploma level. ^#^—those with a previous pregnancy.

**Table 2 nutrients-17-03729-t002:** Descriptive statistics for dietary intake.

	Population Proportion (N)	Population Median (IQR)
Fruit (2 or more per day)	51.7% (215)	2 (1–2)
Vegetables (5 or more per day)	1.7% (7)	1 (1–2)
vegetables (3 or more per day) *	14.7% (61)	1 (1–2)
discretionary (0 per day)	2.9% (12)	1 (0.5–1.5)
discretionary (1 per day)	51.0% (212)	1 (0.5–1.5)
Carbohydrates (8.5 or more per day)	0.7% (3)	3 (2–4)
Carbohydrates (2 or fewer per day)	28% (117)	3 (2–4)
Protein (3.5 or more per day)	0.5% (2)	1 (0.5–1.5)
Protein (2 or more per day)	7.7% (32)	1 (0.5–1.5)
Dairy (3.5 or more per day)	1.4% (6)	1 (0.5–2)
Dairy (2 or more per day)	32.9% (137)	1 (0.5–2)

* Summary of the Australian Dietary Guidelines for Pregnancy [5]: Recommended intake during pregnancy across five food groups: 5 servings of vegetables, 2 servings of fruit, 8 servings of grains (mostly wholegrain), 3.5 servings of lean meat or alternatives, 3.5 servings of dairy or alternatives; also limit intake of the following: Foods high in saturated fat such as many biscuits, cakes, pastries, pies, processed meats, commercial burgers, pizza, fried foods, potato chips, crisps and other savoury snacks; Foods and drinks containing added sugars such as confectionary, sugar-sweetened soft drinks and cordials, fruit drinks, vitamin waters, energy and sports drinks.

**Table 3 nutrients-17-03729-t003:** Participant characteristics associated with favourable/unfavourable composite dietary habits.

Characteristics		Favourable Diet (*n* = 56; 13%)	Unfavourable Diet (*n* = 360; 87%)	*p*
General				
	Median age in years (range)	34 (32–37)	32 (29–36)	0.02
	Median BMI kg/m^2^ (range)	24 (22–29)	25 (22–29)	0.31
Household structure	Any children in household?			
	Yes	80% (43)	69% (246)	0.15
	No	20% (11)	31% (109)	
Educational attainment				
	Incomplete school	0%	3% (10)	0.02
	Complete school	0%	10% (36)	
	School + TAFE	20% (11)	18% (63)	
	School + university	80% (43)	69% (242)	
Financial	Financial autonomy			
	Yes	77% (43)	71% (254)	0.45
	No	18% (10)	25% (89)	
	Unknown	5% (3)	4% (15)	
	household income			
	<50K	2% (1)	6% (21)	0.57
	50–100K	16% (9)	21% (76)	
	100–200K	38% (21)	33% (118)	
	>200K	20% (11)	15% (53)	
	Unknown	25% (14)	25% (87)	
Ethnicity				
	Middle Eastern	21% (12)	19% (67)	0.86
	Other	5% (3)	9% (33)	
	S Asian	39% (22)	36% (131)	
	SE Asian	18% (10)	16% (59)	
	White	16% (9)	19% (70)	

**Table 4 nutrients-17-03729-t004:** Clinical characteristics associated with favourable/unfavourable composite dietary habits.

Characteristic		Favourable Diet (*n* = 56; 13%)	Unfavourable Diet (*n* = 360; 87%)	*p*
Clinical	Median BMI kg/m^2^ (range)	24 (22–29)	25 (22–29)	0.31
	Multiparous			
	Yes	79% (44)	70% (252)	0.21
	No	21% (12)	30% (108)	
	History of GDM			
	Yes	12% (7)	13% (47)	1.00
	No	88% (49)	87% (313)	
	History of PCOS			
	Yes	5% (3)	9% (32)	0.60
	No	95% (53)	91% (328)	
	History of recurrent miscarriage			
	Yes	16% (9)	14% (51)	0.68
	No	84% (47)	86% (309)	
	Assisted conception			
	Yes	7% (4)	5% (17)	0.51
	No	93% (52)	95% (343)	
	High intensity BLISS *			
	Yes	88% (36)	87% (201)	1.00
	No	12% (5)	13% (31)	

* BLISS: Breastfeeding length and intensity scale.

**Table 5 nutrients-17-03729-t005:** Mental health screening scores associated with favourable/unfavourable composite dietary habit.

Mental Health Screening Tool		Favourable Diet (*n* = 56; 13%)	Unfavourable Diet (*n* = 360; 87%)	*p*
DASS21	DASS-Anxiety			
	Normal	85% (40)	75% (189)	0.38
	Mild	4% (2)	9% (23)	
	Moderate+	11% (5)	16% (40)	
	DASS-Depression			
	Normal	94% (44)	92% (232)	1.00
	Mild	4% (2)	5% (13)	
	Moderate+	2% (1)	3% (8)	
	DASS-Stress			
	Normal	74% (35)	67% (170)	0.34
	Mild	2% (1)	9% (22)	
	Moderate+	23% (11)	24% (60)	
EDPS	EPDS			
	Low	88% (49)	90% (322)	0.65
	Moderate	9% (5)	6% (22)	
	High	4% (2)	3% (12)	

**Table 6 nutrients-17-03729-t006:** Physical activity behaviours versus favourable/unfavourable composite dietary habit.

		Favourable Diet (*n* = 56; 13%)	Unfavourable Diet (*n* = 360; 87%)	*p*
Type of physical activity				
	By category			
	Sedentary	11% (6)	17% (60)	0.29
	Insufficient	41% (23)	46% (164)	
	Sufficient	48% (27)	38% (136)	
	Participation in any walking			
	Yes	89% (50)	88% (317)	1.00
	No	11% (6)	12% (43)	
	Participation in any moderate physical activity			
	Yes	21% (12)	17% (63)	0.46
	No	79% (44)	82% (297)	
	Participation in any vigorous physical activity			
	Yes	16% (9)	15% (54)	0.84
	No	84% (47)	85% (306)	
	Participation in any moderate or vigorous physical activity			
	Yes	27% (15)	28% (101)	1.00
	No	73% (41)	72% (259)	

**Table 7 nutrients-17-03729-t007:** Dietary habit and OGTT results (mmol/L).

	Favourable Diet	Unfavourable Diet	*p*
	2+ servings of vegetables and 2+ servings of fruit and 1 discretionary item		
OGTT fasting	4.39 (0.48)	4.43 (0.45)	0.62
OGTT 1 h	7.89 (1.89)	7.92 (2.03)	0.91
OGTT 2 h	6.72 (1.66)	6.71 (1.79)	0.96

**Table 8 nutrients-17-03729-t008:** Rate of gestational diabetes (%) and dietary intake by food group.

Food Group	Yes	No	*p*
Fruit (2 or more per day)	21% (46)	29% (29)	0.09
Vegetables (5 or more per day)	14% (1)	25% (103)	0.69
Vegetables (3 or more per day)	28% (17)	25% (87)	0.63
Discretionary (0 per day)	58% (7)	24% (97)	0.01
Discretionary (1 per day)	27% (57)	23% (47)	0.43
Carbohydrates (8.5 or more per day)	33% (1)	25% (103)	1.00
Carbohydrates (2 or fewer per day)	24% (28)	25% (76)	0.80
Protein (3.5 or more per day)	50% (1)	25% (103)	0.44
Protein (2 or more per day)	34% (11)	24% (93)	0.21
Dairy (3.5 or more per day)	0% (0 out of 6)	25% (104)	0.34
Dairy (2 or more per day)	28% (38)	24% (66)	0.40

## Data Availability

The data presented in this study are available on request from the Chief Principal Investigator and corresponding author (D.P.) due to restrictions as per the published protocol.

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
