# Peer review of "Dietary Habits in Early Pregnancy in a Multi-Ethnic Population: Results from the PROMOTE Cohort Study"

_nutrients, 2025, doi:10.3390/nu17233729_

Round 1

Reviewer 1 Report

Comments and Suggestions for Authors

Thank you for the opportunity of reviewing the manuscript of Samarawickrama et al.

The manuscript presents the results of an observational propsective study investigating the dietary habits in early pregnancy in a multiethnic population od Australia. The Introduction and Discussion sections are well written and the Conclusions are supported by the Results. However the Methods and Results section coul be improved to facilitate understanding. Below are displayed several comments:

  1. Now is unclear which was the objective of the current analysis. Please state it in the abstract and in the Introduction/Methods section.
  2. How were collected data on the social determinants of health/sociodemographic factors and which were them? Please decsribe yjem in the Methods section.
  3. How were the the socio-economic indexes assessed? To be added in the methods section.
  4. How was assessed the physical activity behavior?
  5. What do you understand by preferred intake listed in Table 7. Are these statements referring to any specific intake?

Reviewer 2 Report

Comments and Suggestions for Authors

This paper presents a relatively small-scale study investigating dietary patterns during early pregnancy and their association with gestational diabetes mellitus (GDM) in Australia. The study provides useful background information on the nutritional habits of pregnant women and highlights potential areas of concern in maternal nutrition.

The study design is sound, and the use of standardized questionnaires enhances the reliability of the data collection. The data analysis is straightforward and does not involve complex statistical techniques, which is appropriate given the scope of the study. However, the limited sample size restricts the potential for uncovering novel or statistically significant findings. Future phases of the study, with a larger cohort, may yield more meaningful and generalizable results.

The reference list is current, and the presentation of results is clear and well-organized.

Minor comments:

  • Please check the symbols used in lines 126–127 for accuracy.
  • In Table 1, superscripts should be used where appropriate.

Reviewer 3 Report

Comments and Suggestions for Authors

The manuscript reports findings from the prospective pregnancy cohort study (PROMOTE), conducted in a socially and ethnically diverse population in Western Sydney, Australia. The study focuses on dietary habits during early pregnancy and the possible association with gestational diabetes.

The study design and rationale have been previously detailed by Samarawickrama et al. (BMJ Open 2025; 15(3): e090930). The present work demonstrates an appropriate study design, methodological clarity, and transparent reporting. The results are coherently presented and critically discussed, contributing to the field of maternal nutrition research.

Please standardize the gestational age reported (<16 vs. 15 weeks) to maintain consistency between the abstract and the Methods section. 

Reviewer 4 Report

Comments and Suggestions for Authors

This manuscript reports on dietary habits and their association with Gestational Diabetes Mellitus (GDM) in a multi-ethnic Australian cohort. The study addresses a critical knowledge gap concerning diet in diverse populations during pregnancy and is timely and relevant. The primary strength of the study is its focus on a contemporary, multi-ethnic urban population, a demographic often underrepresented in clinical research. The authors are also to be commended for their transparent reporting of null findings and methodological limitations. However, there are several major concerns must be addressed before this manuscript can be considered for publication. The comments are detailed below.

Major comments

  1. There is a critical and confusing discrepancy in the number of participants classified as having a "favourable diet." The Abstract (line 45) and Results text (line 222) state that n=25 (6%) participants met the pragmatic composite standard. However, all subsequent analyses presented in Tables 2, 3, 4, and 5 use a sample size of n=56 (13%) for the "Favourable diet" group.
  2. The study's primary analysis hinges on a "pragmatic compromise" definition of a favourable diet (≥3 vegetable serves, ≥2 fruit serves, and ≤1 discretionary item). While the authors acknowledge the challenges, the justification for these specific cut-offs is insufficient. Why were these specific thresholds chosen over others? For example, why 3 vegetable serves and not 2 or 4? Is there any precedent in the literature for using this specific composite definition? Citing relevant studies that have used a similar pragmatic approach would significantly strengthen the rationale. Without stronger justification, this definition appears arbitrary and potentially chosen to yield a large enough group for analysis (which, given the n=25/56 discrepancy, is unclear). The authors should provide a more robust defense of this methodological choice in the Methods section (lines 132-136).
  3. In Table 7, the analysis for "Discretionary (0 per day)" shows a GDM rate of 58% versus 24% in the "Not preferred intake" group, with a p-value of 0.01. This is a statistically significant finding. The Results text (lines 274-275) explicitly states, "No statistically significant association was found between individual dietary food groups and rate of gestational diabetes." This is a direct contradiction of the data presented in Table 7. This finding, while likely a statistical artifact due to the very small cell size (n=12 in the "0 discretionary" group, of which 7 had GDM), is highly counter-intuitive and cannot be ignored. The authors must address this result in the text. They should discuss the possibility of it being a Type I error due to multiple comparisons and small sample size, or explore potential explanations (e.g., reverse causality, where high-risk women are attempting extreme dietary changes). Simply omitting it from the narrative is not acceptable.
  4. The authors correctly identify the use of a short food frequency questionnaire (FFQ) as a limitation. However, for a manuscript submitted to a specialized journal like Nutrients, this point requires a more in-depth discussion. The 13-item questionnaire is a very crude instrument. The authors should elaborate on what specific dietary information is lost with this tool (e.g., distinction between wholegrain and refined carbohydrates, types of fats, portion size accuracy, preparation methods). This limitation directly impacts the study's ability to draw meaningful conclusions about "diet quality." The discussion should more strongly temper the conclusions by emphasizing that the null findings could be a result of measurement error from the blunt instrument used, rather than a true lack of association.
  5. The title and abstract frame the study around the association between diet and GDM. However, the study was severely underpowered to investigate this relationship, a fact the authors acknowledge. The most robust and valuable finding of this paper is the descriptive epidemiology of dietary habits—specifically, the extremely low adherence to national guidelines in this diverse population. I suggest the authors consider reframing the Title and Abstract to highlight this as the primary finding.

Minor comments

Abstract. The statement "Association between dietary habits and GDM was difficult to assess" (lines 51-52) is an understatement. It would be more accurate to state that the study was underpowered to detect an association and the analysis was inconclusive due to the highly skewed distribution of dietary patterns.

Methods (Line 144). The ADIPS criteria are cited. It would be helpful for an international audience to briefly state the glucose thresholds in the text, as is done later in lines 146-147. This is a minor point of clarity.

Discussion (Line 325). The statement "Our study was unable to meaningfully calculate a sample size" is slightly confusing. It seems the authors mean a post-hoc power calculation was not meaningful, or that a prospective sample size calculation for a future study is challenging. Please clarify.

Round 2

Reviewer 1 Report

Comments and Suggestions for Authors

Thank you for the opportunity to review the revised version of the manuscript. All the comments have been appropriately addressed. I have no further comments,

Reviewer 4 Report

Comments and Suggestions for Authors

Can be accepted.